# Comparison of Longitudinal and Cross-Sectional Approaches in Studies on Knowledge, Attitude and Practices Related to Non-Medical Tranquilizer Use

**DOI:** 10.3390/jcm10214827

**Published:** 2021-10-21

**Authors:** Narmeen Mallah, Julia Battaglia, Adolfo Figueiras, Bahi Takkouche

**Affiliations:** 1Department of Preventive Medicine, University of Santiago de Compostela, 15782 Santiago de Compostela, Spain; narmeen.mallah@usc.es (N.M.); julia.battaglia@rai.usc.es (J.B.); adolfo.figueiras@usc.es (A.F.); 2Centro de Investigación Biomédica en Red de Epidemiología y Salud Pública (CIBER-ESP), 28029 Madrid, Spain; 3Health Research Institute of Santiago de Compostela (IDIS), 15706 Santiago de Compostela, Spain

**Keywords:** cross-sectional study, cohort study, knowledge-attitude-practice, non-medical tranquilizer use, Spain

## Abstract

Research about the association of knowledge and attitudes with practices (KAP) of non-medical tranquilizer use is scarce. We compared findings from cross-sectional and longitudinal approaches in a KAP-based study on non-medical tranquilizer use in Spain using data collected from the same population. Eight-hundred forty-seven participants completed a validated KAP questionnaire at baseline and were then followed-up bimonthly for one year for episodes of non-medical tranquilizer use. Non-medical use was defined as unprescribed use, non-adherence to treatment, storage/sharing of tranquilizers, or a combination of those practices. Adjusted odds ratios (ORs) and their 95% confidence intervals (CIs) were estimated using logistic regression from cross-sectional data and generalized linear mixed models for repeated measures in the longitudinal approach. Only the longitudinal approach showed that limited knowledge about the effect of tranquilizers on behaviour [OR: 3.24 (95% CI: 1.12–9.38)] and about the negative effect of their excessive consumption [OR: 4.12 (95% CI: 1.5–11.33)] is associated with storing/sharing tranquilizers. Both cross-sectional and longitudinal analyses indicated that personal attitudes towards tranquilizers and attitudes towards healthcare providers are associated with non-medical tranquilizer use, yet with different magnitude of associations. Differences between the two approaches were also observed for individual types of non-medical use. Certain discrepancies exist between findings from longitudinal and cross-sectional approaches on KAP of non-medical tranquilizer use. KAP studies are the backbone for designing and evaluating prevention programs on non-medical tranquilizer use, and hence choosing a proper study design, scrutinizing the associated biases, and carefully interpreting findings from those studies are required.

## 1. Introduction

Non-medical tranquilizer use is an internationally growing public health concern with heavy economic and social consequences [1]. The non-medical use practices entail the intake of tranquilizers without medical prescription, non-adhering to physician indications of use, and/or storage or sharing of tranquilizer leftover [2]. Self-medication with psychoactive drugs can take place by using old prescriptions or sharing those drugs with other individuals [3,4]. Non-adherence to tranquilizer treatment occurs when taking the drug in a dose higher or lower than prescribed and for a shorter or longer period than recommended [5,6,7]. Non-medical tranquilizer use is associated with serious health consequences, complicated management of comorbid diseases, increased risk of motor vehicle collision, worsened quality of life, and thus, increments the rates of hospitalization, healthcare expenses and mortality [6,7,8,9].

The frequency of non-medical tranquilizer use among adolescents and adults in the United States is as high as that of cocaine consumption [1,9], and the mortality rate attributed to drug overdose including tranquilizers incremented 10 folds over the past decade in the country [10]. One-tenth of the European population declared having ever non-medically used tranquilizers [11].

Non-medical tranquilizer use is also frequent in countries with lower socioeconomic status [12,13,14,15].

Few studies have been performed to identify the determinants of non-medical tranquilizer use, most of which examined sociodemographic determinants such as age [1,16], gender [17,18], and education [19,20]. Other studies determined the association of non-medical tranquilizer use with co-ingestion of other substances [21], impulsivity [22,23] and antecedents of psychological, physical, and medical conditions [1]. The most common reported motives for non-medical tranquilizer use are to aid or induce sleep, help control anxiety, relax, experiment new feelings, or to use tranquilizers for recreational motives [24,25,26]. Other reported motives of non-medical tranquilizer use include potentiating the effect of illicit drugs, self-harm, and suicide [26].

Knowledge, attitude and practice (KAP) studies about non-medical tranquilizer use are scarce. Only two recent studies showed that knowledge about tranquilizers, personal attitudes towards those drugs and attitudes towards the healthcare provider are associated with non-medical tranquilizer use [27,28].

Literature about other medicines revealed that KAP based studies are widely carried out using cross-sectional designs. Likewise, for KAP studies about non-medical tranquilizer use, it is predicted that in many settings, cross-sectional designs will be preferred over longitudinal ones due to their simplicity, feasibility, rapidity, and cost efficiency. Knowing that cross-sectional designs are less adapted to infer causal relationships [29], we aimed in this study to compare the findings of cross-sectional data collected from the same individuals at baseline in a longitudinal study using a validated KAP questionnaire about tranquilizer use [30]. Our objective is to verify to which extent cross-sectional analyses on KAP of non-medical tranquilizer use yield valid results, similar to those of follow-up data, at a lower cost.

## 2. Methods

### 2.1. Study Population and Setting

Adult children’s caregivers (≥18 years old) visiting the primary care settings of the University Hospital of Santiago de Compostela in 2019 were recruited for this study. Santiago de Compostela is the administrative capital of Galicia, Spain. It has 80,000 inhabitants and a university of 25,000 students.

A minimum sample size of 686 individuals was determined by considering two-sided alpha level = 0.05, statistical power = 80%, a ratio of unexposed/exposed individuals = 7 and an odds ratio (OR) = 2.

Each participant was asked to complete a self-administered questionnaire about knowledge, attitude, and practices of tranquilizer use. The questionnaire was previously validated in the same population (i.e., Galician/Spanish adult population [30]). The baseline questionnaire consisted of 35 questions: items 1 to 16 inquired about knowledge and attitudes regarding tranquilizers. The attitude items evaluated two concepts: personal attitudes towards tranquilizers and patients’ attitudes towards healthcare providers. Participants expressed their level of agreement with each of these 16 knowledge and attitude statements using a scale of 0 (strongly disagree) to 10 (strongly agree). Questions 1 to 4 and 11 and 13 explored personal attitudes towards tranquilizers. Questions 5 to 8 and 10 determined participants’ knowledge about tranquilizers. Questions 9 and 12 to 16 investigated patients’ attitudes towards healthcare provider [30]. The item Q13 shares content with the two attitude factors (attitude toward tranquilizers and attitude toward healthcare providers) [30]. Questions 17 to 25 established whether the participants had taken tranquilizers in the past two months and if non-medical use had occurred. Participants were asked about the source of tranquilizers, i.e., medical prescription or other sources (Q19), their adherence with the indicated course of treatment in terms of duration (Q20) and dosage (Q22–25), and the action taken regarding any tranquilizer leftover (Q21). Questions 28 to 35 assessed the participants’ sociodemographic characteristics including gender, age, educational attainment, working status, family size, alcohol drinking habits, frequency of consulting a doctor in case of sickness and ever receiving any medical prescription over the phone.

As general knowledge and opinions regarding the attitudes are not conditioned to tranquilizer use, participants were instructed to answer those statements whether they were using tranquilizers or not. Those who declared not using tranquilizers in the past two months or who used tranquilizers with no sign of non-medical use were considered as not presenting the outcome.

Participants were followed-up every two months by phone to inquire about their use of tranquilizers. During the follow-up, questions of the practice block (Q17–Q25) were asked.

### 2.2. Exposure

The exposure was defined as lack of knowledge about tranquilizers or the existence of medically inappropriate attitudes. It was measured using the items Q1–Q16, which were analysed separately (i.e., 16 exposures were ascertained in the present study).

### 2.3. Outcome

Five outcomes were defined: (1) use without medical prescription, (2) shortening the course of treatment, (3) sharing or storage of tranquilizer leftover, (4) modifying the prescribed dose, and (5) incorrect action when skipping a previous dose (i.e., doubling the dose or taking it when remembered). In addition, a sixth composite outcome “any non-medical use” was defined and consisted of any one or any combination of the previously mentioned five outcomes.

### 2.4. Statistical Analysis

Two different approaches were undertaken, namely cross-sectional and longitudinal.

Cross-sectional approach: data were obtained from the baseline questionnaire of the longitudinal study and analysed using multivariate logistic regression models.

Longitudinal approach: data were generated from follow-up assessments of the subjects and analysed using multivariate generalized linear mixed models (GLMM). As the outcome, non-medical tranquilizer use, is a binary variable, GLMM models were fitted with the binomial family.

In both approaches, each of the 16 knowledge and attitude items represented an independent variable. The items were categorized into tertiles of the distribution of the outcome. The category that represents the highest level of knowledge or the most medically approved attitude was chosen as a referent. The estimated ORs and their 95% confidence intervals (CIs) were adjusted for age and gender. In addition, in a univariate analysis, we tested the other potentially confounding sociodemographic variables. Variables that showed a *p*-value < 0.2 were introduced successively into the multivariate model, and those that modified the OR previously adjusted for age and gender by 10% or more were included in the final model [31]. Only outcomes of sufficient observations at baseline and during the follow-up were evaluated. To allow for causal inference, the longitudinal data did not include baseline outcomes.

The generated and analysed datasets are available in the data repository FigShare (see Data Availability section). Analyses were carried out using SPSS (SPSS Inc. Released 2011. SPSS for Windows, Version 20.0. Chicago, IBM, Armonk, NY, USA), and mgcv package of R Statistical Software (version 4.0.5) (R Core Team, Vienna, Austria) [32].

## 3. Results

### 3.1. Study Population

Eight-hundred ninety people were approached and asked for participation in the study, of whom 847 answered the baseline questionnaire and 747 completed at least on follow-up assessment. The age of the participants ranged between 18 and 76 years, with a median of 42. Most of the participants were females (75.4%), employed (75.3%), with high academic level (62.9%) and belonging to households of four family members or less (81.1%) (Table 1).

At baseline, 75 (8.9%) participants showed at least one aspect of non-medical tranquilizer use (any non-medical use) in the two months preceding the study. During follow-up, 124 (9.2%) events of any non-medical use practices of tranquilizers were reported. Table 2 summarizes the frequency of specific aspects of non-medical tranquilizer use at baseline and during the follow-up.

Given the number of observations per type of non-medical tranquilizer use, we were able to analyse the association of knowledge and attitudes with the following practices: any non-medical use, shortened treatment, sharing or storing tranquilizer leftover, and modification of the prescribed dose.

### 3.2. Association of Knowledge with Practices of Non-Medical Tranquilizer Use

Both approaches, cross-sectional and longitudinal, showed that overall, the level of knowledge of the population about tranquilizers is not associated with their non-medical use (Q5–Q8 and Q10) (Appendix A). Nevertheless, when analysing each type of non-medical tranquilizer use separately, the longitudinal approach showed that not knowing that tranquilizers can affect people´s control over what they do (Q5) or ignoring that an excessive consumption of tranquilizers might reduce their effect in the future (Q10), are associated with increased odds of “sharing or storing tranquilizer leftover” [aOR_1st tertile_ = 3.24 (95% CI: 1.12–9.38) and aOR_1st tertile_ = 4.12 (95% CI: 1.5–11.33), respectively]. These associations were not observed in the cross-sectional approach (Appendix A).

### 3.3. Association of Personal Attitude towards Tranquilizers with Practices of Non-Medical Tranquilizer Use

The cross-sectional and longitudinal approaches revealed a substantial association between personal attitudes towards tranquilizers (Q1–Q4, Q11 and Q13) and their non-medical use practices, yet the magnitude of association was different (Appendix A).

### 3.4. Association of Patients’ Attitudes towards Healthcare Provider and Practices of Non-Medical Tranquilizer Use

There are discrepancies in the results of the two approaches regarding the association of patients´ Attitudes towards healthcare provider (Q9 and Q12–16) with non-medical tranquilizer use. In particular, item Q12 (incomplete trust with the doctors’ decision to prescribe tranquilizers or not) was not associated with the outcome “any non-medical tranquilizer use practice” in the cross-sectional approach, contradicting the results of the longitudinal approach (Appendix A). Nonetheless, item Q12 was associated with higher odds of the outcomes “shortening the course of treatment” and “modifying the prescribed dose”, whereas in the longitudinal approach, no association was observed between Q12 and “modifying the prescribed dose” (Appendix A).

The outcome “shortening the course of treatment” could not be analysed longitudinally due to the limited number of observations.

## 4. Discussion

Research about the association of knowledge and attitudes with practices of non-medical tranquilizer use is recent and to the best of our knowledge only two studies have addressed this issue so far [27,28].

In the literature about other medicines, KAP based studies using a cross-sectional design are highly abundant as this design requires less resources and time than a longitudinal one. For instance, out of thousands of studies about KAP on non-medical antibiotic use, none was longitudinal until the publication of a recent comparative study between the two approaches [33]. The main disadvantage of cross-sectional studies is that since exposure and outcome are measured at the same time, simultaneity bias is a concern as it cannot be assured whether the exposure preceded the outcome or if the inadequate knowledge or attitude was acquired at the time of the non-medical use [29]. Moreover, cross-sectional and longitudinal designs evaluate different epidemiological concepts and readers could misinterpret a prevalence odds ratio of cross-sectional studies as a proxy of relative risk determined by longitudinal studies [34].

Cross-sectional and longitudinal designs are expected to generate different conclusions [33,35,36,37,38], and cross-sectional approaches are often deemed misleading, yet in the context of KAP based studies, biases associated with both designs should be inspected [33].

In the current study, cross-sectional and longitudinal approaches yielded similar conclusions with respect to the association of knowledge and personal attitudes towards tranquilizers with the outcome “any non-medical use” practices, though the estimated magnitude of association moderately differed between the two designs. However, differences were observed upon analysing each type of non-medical tranquilizer use separately. Moreover, regarding the association of patients’ attitudes towards healthcare providers (Q9, Q12–16), findings from the cross-sectional and the longitudinal approaches were divergent.

Various sources of biases could have contributed to the different findings from cross-sectional and longitudinal approaches. During phone follow-up assessment, participants are less likely to admit their medically disapproved attitudes than during a self-administered questionnaire, which results in a fewer number of outcomes during the follow-up [39]. Nonetheless, in our settings, in both approaches, a sufficient number of observations was obtained for several outcomes, except for “shortening the course of treatment” where few observations were obtained during the follow-up.

Selection bias due to loss of follow-up is another concern. Participants with a lower level of knowledge are more likely to abandon follow-up due to poorer adherence to treatment [40].

Nevertheless, in our study, the level of knowledge and attitudes of participants lost to follow-up was similar to those included in the study. Hence, selection bias is unlikely to affect our results.

Given that non-medical tranquilizer use is an increasingly common public health issue with significant consequences, it is predictable that numerous studies around the world will start investigating the psychosocial determinants of non-medical use in their settings. Intervention studies based on KAP assessments are also foreseen to take place to help assess and design prevention programs to improve the rationale use of tranquilizers.

This study provided a methodological assessment of cross-sectional and longitudinal designs to evaluate the associations of knowledge and attitudes with practices of non-medical tranquilizer use. In our settings, findings from the two approaches were not consistent for the association of certain knowledge and attitude statements with non-medical tranquilizer use in general, as well as with specific types of non-medical use. Fundamental methodological aspects were stressed and should prove useful for future studies. This may help researchers choose a design and adequately interpret findings generated from cross-sectional and longitudinal KAP studies in the specific case of tranquilizers.

## Figures and Tables

**Table 1 jcm-10-04827-t001:** Sociodemographic characteristics of the study population.

Characteristic	Total (*N* = 847)	Non-Medical Use (*N* = 75)	No Non-Medical Use (*N* = 772)
**Sex**			
Male	208 (24.6%)	9 (12.0%)	199 (25.8%)
Female	639 (75.4%)	66 (88.0%)	573 (74.2%)
Missing	0	0	0
**Age (years)**			
≤35	149 (17.6%)	10 (13.3%)	139 (18%)
36–45	425 (50.2%)	38 (50.7%)	387 (50.1%)
≥46	273 (32.2%)	27 (36.0%)	246 (31.9%)
Missing	0	0	0
**Educational level**			
Until high school	285 (33.6%)	32 (42.7%)	253 (32.8%)
University	533 (62.9%)	43 (57.3%)	490 (63.5%)
Missing	29 (3.4%)	0	29 (3.8%)
**Family size**			
≤4	687 (81.1%)	68 (90.7%)	619 (80.2%)
>4	129 (15.2%)	7 (9.3%)	122 (15.8%)
Missing	31 (3.7%)	0	31 (4.0%)
**Consulting a doctor**			
Not always	434 (51.2%)	38 (50.7%)	396 (51.3%)
Always	381 (45%)	37 (49.3%)	344 (44.6%)
Missing	32 (3.8%)	0	32 (4.1%)
**Medical consultation over the phone**			
No	471 (55.6%)	32 (42.7%)	439 (56.9%)
Yes	342 (40.4%)	42 (56.0%)	300 (38.9%)
Missing	34 (4%)	1 (1.3%)	33 (4.3%)
**Employment status**			
Employed	638 (75.3%)	53 (70.7%)	585 (75.8%)
Unemployed	177 (20.9%)	21 (28.0%)	156 (20.2%)
Missing	32 (3.8%)	1 (1.3%)	31 (4.0%)
**Alcohol Intake**			
Never/less than once per month	479 (56.6%)	53 (70.7%)	426 (55.2%)
Others	336 (39.7%)	22 (29.3%)	314 (40.7%)
Missing	32 (3.8%)	0	32 (4.1%)
**Sex**			
Male	208 (24.6%)	9 (12.0%)	199 (25.8%)
Female	639 (75.4%)	66 (88.0%)	573 (74.2%)
Missing	0	0	0
**Age (years)**			
≤35	149 (17.6%)	10 (13.3%)	139 (18%)
36–45	425 (50.2%)	38 (50.7%)	387 (50.1%)
≥46	273 (32.2%)	27 (36.0%)	246 (31.9%)
Missing	0	0	0
**Educational level**			
Until high school	285 (33.6%)	32 (42.7%)	253 (32.8%)
University	533 (62.9%)	43 (57.3%)	490 (63.5%)
Missing	29 (3.4%)	0	29 (3.8%)
**Family size**			
≤4	687 (81.1%)	68 (90.7%)	619 (80.2%)
>4	129 (15.2%)	7 (9.3%)	122 (15.8%)
Missing	31 (3.7%)	0	31 (4.0%)
**Consulting a doctor**			
Not always	434 (51.2%)	38 (50.7%)	396 (51.3%)
Always	381 (45%)	37 (49.3%)	344 (44.6%)
Missing	32 (3.8%)	0	32 (4.1%)
**Medical consultation over the phone**			
No	471 (55.6%)	32 (42.7%)	439 (56.9%)
Yes	342 (40.4%)	42 (56.0%)	300 (38.9%)
Missing	34 (4%)	1 (1.3%)	33 (4.3%)
**Employment status**			
Employed	638 (75.3%)	53 (70.7%)	585 (75.8%)
Unemployed	177 (20.9%)	21 (28.0%)	156 (20.2%)
Missing	32 (3.8%)	1 (1.3%)	31 (4.0%)
**Alcohol Intake**			
Never/less than once per month	479 (56.6%)	53 (70.7%)	426 (55.2%)
Others	336 (39.7%)	22 (29.3%)	314 (40.7%)
Missing	32 (3.8%)	0	32 (4.1%)
**Sex**			
Male	208 (24.6%)	9 (12.0%)	199 (25.8%)
Female	639 (75.4%)	66 (88.0%)	573 (74.2%)
Missing	0	0	0
**Age (years)**			
≤35	149 (17.6%)	10 (13.3%)	139 (18%)
36–45	425 (50.2%)	38 (50.7%)	387 (50.1%)
≥46	273 (32.2%)	27 (36.0%)	246 (31.9%)
Missing	0	0	0
**Educational level**			
Until high school	285 (33.6%)	32 (42.7%)	253 (32.8%)
University	533 (62.9%)	43 (57.3%)	490 (63.5%)
Missing	29 (3.4%)	0	29 (3.8%)
**Family size**			
≤4	687 (81.1%)	68 (90.7%)	619 (80.2%)
>4	129 (15.2%)	7 (9.3%)	122 (15.8%)
Missing	31 (3.7%)	0	31 (4.0%)
**Consulting a doctor**			
Not always	434 (51.2%)	38 (50.7%)	396 (51.3%)
Always	381 (45%)	37 (49.3%)	344 (44.6%)
Missing	32 (3.8%)	0	32 (4.1%)
**Medical consultation over the phone**			
No	471 (55.6%)	32 (42.7%)	439 (56.9%)
Yes	342 (40.4%)	42 (56.0%)	300 (38.9%)
Missing	34 (4%)	1 (1.3%)	33 (4.3%)
**Employment status**			
Employed	638 (75.3%)	53 (70.7%)	585 (75.8%)
Unemployed	177 (20.9%)	21 (28.0%)	156 (20.2%)
Missing	32 (3.8%)	1 (1.3%)	31 (4.0%)
**Alcohol Intake**			
Never/less than once per month	479 (56.6%)	53 (70.7%)	426 (55.2%)
Others	336 (39.7%)	22 (29.3%)	314 (40.7%)
Missing	32 (3.8%)	0	32 (4.1%)

**Table 2 jcm-10-04827-t002:** Frequency of reported types of non-medical tranquilizer use at baseline and during the follow-up.

Type of Non-Medical Tranquilizer Use	Cross-Sectional Approach (Baseline Data, *N* = 847)	Longitudinal Approach (Follow-Up Data, *N* = 1343)
Any non-medical use	75 (8.9%)	124 (9.2%)
Use without prescription	8 (0.9%)	60 (4.5%)
Shortening the course of treatment	25 (3.0%)	16 (1.2%)
Sharing or storage of tranquilizer leftover	48 (5.7%)	57 (4.2%)
Modifying the prescribed dose	34 (4.0%)	39 (2.9%)
Doubling the dose or taking it when remembered, when skipping a previous dose	9 (1.1%)	26 (1.9%)

## Data Availability

The data generated and analyzed in this study is available in the data Repository FigShare [DOI:10.6084/m9.figshare.14526954].

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
