# Peer review of "Comparison of Longitudinal and Cross-Sectional Approaches in Studies on Knowledge, Attitude and Practices Related to Non-Medical Tranquilizer Use"

_jcm, 2021, doi:10.3390/jcm10214827_

Round 1
Reviewer 1 Report
Interesting paper. But two major issues:
(1) That XS and longitudinal models give different results is already well known.
(2) A comparison of XS and longitudinal generalized linear models (e.g., logistic regression) is incomplete without discussion of they type of longitudinal model being used - a random effects (conditional, individual-level) model (GLMM) or a population-averaged model (GEE).
You begin by stating "Research about the association of Knowledge and Attitudes with Practices (KAP) of tranquilizer misuse is scarce." But then go on to compare XS and longitudinal approaches. The fact that XS and longitudinal models can give different results is well-known. Why not make the focus of the paper on the association itself? For example, with longitudinal data you could model the change in the response over time and see if any of the K or A variables are associated with the intercept or slope (Y ~ Intercept + Time + X + X:Time). Or you could just use the longitudinal model as you have it (if change over time is not of interest) to answer the question. There does not seem to be any reason to use the XS data.
Consider if changing the focus from (XS vs. Longitudinal) to (What is the association between K,A and P?) makes more sense and would provide a more meaningful contribution to the literature. That being said, just comparing the XS and longitudinal approaches in this setting to illustrate what is known is still OK. But given your opening sentence, I would still consider making the focus the association itself.
"Those who declared not using tranquilizers in the past two months or who used tranquilizers with no sign of misuse, served as a comparison group."
This is confusing and implies that this is the exposed vs. not exposed comparison. But use with misuse is your outcome (for the first analysis).
In your GLMM model, specify what random effects you included. Was it just a random intercept or was the exposure effect also included as a random effect? Did you include time in the model (fixed or random)?
Some discussion needs to be added regarding why the cross-sectional model and longitudinal model differ. They are in fact estimating different things and this is part of the explanation for why XS effects differ from the longitudinal effects.
When fitting a generalized linear model, you can fit either a marginal model (using GEE) or a conditional model (using random effects). Unlike with a linear model, the interpretation of these two is not the same in a generalized linear model (like logistic regression).
Your cross-sectional (baseline data only) model estimates the marginal (population-averaged) effect of exposure on outcome. The corresponding longitudinal model would be a logistic regression fit using generalized estimating equations (GEE). The longitudinal model you fit is a logistic regression with random effects which estimates a conditional (individual-level) effect.
For example, if the outcome is "use without medical prescription":
In a conditional (random effects) model, the OR represents the effect of the exposure on an individual person's probability of use without medical prescription.
In a marginal (cross-sectional or longitudinal GEE) model, the OR represents the difference in prevalence of use without medical prescription between groups of individuals who differ in exposure.
*** In general, the population-average effect will be smaller than the individual-level effect. ***
In theory, using GEE will give you results that are similar to the cross-sectional model, but with more precision since you have more data in the longitudinal case.
If you want to compare the XS and longitudinal approaches, you have to clarify what it is you are trying to compare. What you are currently showing is that a XS approach (which is, by necessity, a population-average approach) gives different results than an individual-level longitudinal approach. Again, this is not new information.
(An excellent text on this topic is "Analysis of Longitudinal Data" by Diggle, Heagerty, Liang, and Zeger.)
How were the 890 people you approached selected from the population? Specify. Sounds like you have a convenience sample? Add to the Discussion a statement about how your sampling method results in a non-representative sample and your results may be biased.
Add the # of responses at each follow-up.
Clarify which of Q1 - Q16 are K and which are A. That would help in reading the Results.
Table 1. Add a column for "No Misuse".
Clarify the levels of "Educational level". You could have <HS, HS but no university, some university but not graduated, graduated from university. What do your levels include?
Clarify "Family size". Who does this include?
Table 2. The header makes it sound like your Longitudinal sample of 1343 is just follow up data, not the baseline data. That makes sense since that way the exposures (which were measured only at baseline) all precede your outcome. If you actually included the baseline outcomes in the longitudinal dataset, take them out so your exposures all precede your outcomes. Clarify in your Methods that the longitudinal data does not include baseline outcomes, and that that allows causal inferences.
"Variables that showed a p-value <0.2 were introduced successively into the multivariate model, and those that modified the OR previously adjusted for age and gender by 10% or more were included in the final model (Greenland, 1989). Only outcomes of sufficient observations at baseline and during the follow-up were evaluated."
Using p-values to decide which confounders to include in your adjusted model biases your final model p-value. Instead, decide ahead of time which potential confounders to include and include them regardless of significance or magnitude of OR. See the text "Regression Modeling Strategies" by Frank Harrell, Jr. for more information.
Greenland 1989 which you cite states, "variable-selection algorithms in current packaged programs, such as conventional stepwise regression, can easily lead to invalid estimates and tests of effect". What you did (p < .2 as your p to enter) is essentially the same thing as a stepwise procedure. But even the 10% rule can be problematic. Deciding what confounders to include based on the strength or magnitude of association of the confounders with the outcome leads to invalid p-values. Instead, decide a priori what variables may be confounders (excluding those that may be mediators) and include them all in the model, with the caveat that you should not have more than m/10 predictors in the model, where m = min(# with Y = 1, # with Y = 0). (Harrell, p73).
You carried out a large number of tests and did not adjust for multiple testing. Thus, your Type I error rate is far larger than 0.05. In the Limitations, note that you did not adjust for multiple testing and these results should be considered exploratory (hypothesis generating), not confirmatory. If in future for a different study you want to do a confirmatory analysis, specify a set of primary tests a priori and adjust for multiple testing over those tests, with everything else considered secondary. But you can't choose based on the p-values... you would have had to choose ahead of time. So for this analysis you are limited at this point to keeping this as exploratory.
I found the Results section difficult to read. I went into it thinking your main point was comparing XS and longitudinal effects, but you mix between highlighing interesting associations and comparing these effects.
"Moreover, cross-sectional and longitudinal designs evaluate different epidemiological concepts and readers could misinterpret a prevalence odds ratio of cross-sectional studies as a proxy of relative risk determined by longitudinal studies (Rothman, 1986)."
But you did not compute relative risk in the longitudinal model.
"Nevertheless, in our study, the level of Knowledge and Attitudes of participants lost to follow-up was similar to those included in the study"
This is not clear. What was your response rate at each follow-up?
Author Response
Dear Dr. Lodi,
We would like to thank you and the reviewers for your recommendations concerning our manuscript "Comparison of longitudinal and cross-sectional approaches in studies on Knowledge, Attitude and Practices related to tranquilizer misuse” (jcm-1229883).
Below we provide our response to each of the comments made by the reviewers. We also provide a version of the article in which all changes are highlighted.
We hope that the amended paper will now be deemed acceptable for publication in the Journal of Clinical Medicine.
Sincerely,
Bahi Takkouche MD, Ph.D
Professor of Epidemiology
University of Santiago de Compostela
Spain
Editor:
Comment: "Increasingly the concept "misuse" is interpreted as a stigmatizing
terminology, and the authors should consider if they could use an
alternative phrase, such as "non-medical use".
Answer: We have modified the term “misuse” as suggested, throughout the paper.
Comment: It is not clear to me how the authors have dealt with the fact that they
have examined the respondents multiple times in the longitudinal vs one
time for the cross-sectional design in terms of the time available for
exposure to misuse practices.
Answer: In the cross-sectional approach (from now on called CSA), by definition, follow-up time is not taken into account. This is why we used GLMM in the longitudinal approach (from now on called LA) but logistic regression in the CSA. For the latter, we assessed the relation between exposure and outcome at baseline, i.e. at enrollment. Outcome events (non-medical use of tranquilizers) are prevalent i.e. they already exist at enrollment in the study.
Comment: So their reported outcome has been linked to the data points and the
observation period "the prior two months" to each interview as I understand it. In the cross-sectional design they included one such period, while for the longitudinal design, multiple such periods, allowing their respondents to be exposed for the risk of "misuse" for a much longer time all data combined, than in the cross-sectional design.
Answer: As explained in the answer to the previous comment, as per classic epidemiologic definition, the CSA does not include time of observation into account. Exposure and outcome are measured at the same time, and their relation assessed without using time of observation. There is nothing awkward in this approach, except the drawback inherent to cross-sectional studies: the difficulty in assessing causation as the time sequence is not defined, i.e. it is not known whether exposure (knowledge and attitude) preceded or not the occurrence of the outcome (non-medical use).
Comment: Thus prevalence estimates of specific misuse patterns, will necessarily be expected to be higher from the longitudinal design if they have not addressed the total time available for observing misuse."
Answer: In GLMM, the follow-up time is considered by the fact that each time a participant completes a follow-up assessment, the corresponding data is registered as a separate observation. Therefore, we believe that the prevalence is not forcedly higher in the longitudinal design.
Reviewer #1:
Interesting paper. But two major issues:
Comment: That XS and longitudinal models give different results is already well known.
Answer: To the best of our knowledge, we were the first to design and validate a KAP questionnaire on tranquilizer non-medical use and apply it in cross-sectional and longitudinal approaches. Therefore, this study represents the first comparison between the two designs in any topic related to ours. We would have been grateful to have a list of references of articles tackling this issue. In spite of an intensive search, we did not find any article comparing the two approaches.
Comment: A comparison of XS and longitudinal generalized linear models (e.g., logistic regression) is incomplete without discussion of they type of longitudinal model being used - a random effects (conditional, individual-level) model (GLMM) or a population-averaged model (GEE).
Answer: We would like to clarify that in the manuscript we informed that our longitudinal approach data were analyzed using multivariate GLMM models. These models allow for the introduction of random terms to control initial intra-individual heterogeneity. To construct the models, we used tranquilizer misuse as the dependent variable, with individual observations (per questionnaire and participant) as level 1 and participants as level 2. Random intercept effects were considered among participants. Also, we used the binomial family to fit GLMM models as the outcome, was a binary variable.
Comment: You begin by stating "Research about the association of Knowledge and Attitudes with Practices (KAP) of tranquilizer misuse is scarce." But then go on to compare XS and longitudinal approaches. The fact that XS and longitudinal models can give different results is well-known. Why not make the focus of the paper on the association itself? For example, with longitudinal data you could model the change in the response over time and see if any of the K or A variables are associated with the intercept or slope (Y ~ Intercept + Time + X + X:Time). Or you could just use the longitudinal model as you have it (if change over time is not of interest) to answer the question. There does not seem to be any reason to use the XS data.
Answer: We thank the reviewer for this comment; however, we believe that doing one thing (assessing the relation between KAP and misuse in a longitudinal study) is not incompatible with the other (comparing results from the two approaches). The aim of this paper was clearly stated from the beginning: to compare the two approaches. Our rationale was that, often, cross-sectional studies are carried out without paying much attention to methodological details, and their results are interpreted the same way as if they were longitudinal studies. This is occurring frequently, especially in KAP studies. For instance, out of numerous KAP studies on antibiotic misuse we were the first to conduct a longitudinal approach and to highlight methodological aspects associated with each design (Mallah et al. Comparison of longitudinal and cross-sectional approaches in studies about knowledge, attitude, and practices related to antibiotic misuse. Drug Saf. 2021 44(7):797-809) As epidemiologists who care about the sound interpretation of results, we believe it is interesting to discuss methodological approaches without much technical detail in a journal such as JCM, read by clinicians and physicians. This did not prevent us from publishing recently, in a major journal, an article entirely devoted to the cohort study (Mallah et al. Association of knowledge and attitudes with practices of misuse of tranquilizers: A cohort study in Spain. Drug Alcohol Depend 2021; 225:108793)
Comment: Consider if changing the focus from (XS vs. Longitudinal) to (What is the association between K,A and P?) makes more sense and would provide a more meaningful contribution to the literature. That being said, just comparing the XS and longitudinal approaches in this setting to illustrate what is known is still OK. But given your opening sentence, I would still consider making the focus the association itself.
Answer: We thank the reviewer for this suggestion, but as previously mentioned we have already published the association between Knowledge, Attitudes, and tranquilizer misuse Practices (Mallah et al. Association of knowledge and attitudes with practices of misuse of tranquilizers: A cohort study in Spain. Drug Alcohol Depend 2021; 225:108793). Nonetheless, knowing that cross-sectional design is widely adopted in studies on the misuse of other drugs such as antibiotics due to the rapidity and low cost of this design, we found that it is essential to compare findings from cross-sectional and longitudinal approaches in the context of the KAP of tranquilizer non-medical use and highlight on important methodological issues.
Comment: "Those who declared not using tranquilizers in the past two months or who used tranquilizers with no sign of misuse, served as a comparison group." This is confusing and implies that this is the exposed vs. not exposed comparison. But use with misuse is your outcome (for the first analysis).
Answer: We agree. We have modified the writing as follows: “Those who declared not using tranquilizers in the past two months or who used tranquilizers with no sign of non-medical use were considered as not presenting the outcome.”
Comment: In your GLMM model, specify what random effects you included. Was it just a random intercept or was the exposure effect also included as a random effect? Did you include time in the model (fixed or random)?
Answer: As explained earlier, we used random intercept effects. Time, as an independent variable, was not included in the model.
Comment: Some discussion needs to be added regarding why the cross-sectional model and longitudinal model differ. They are in fact estimating different things and this is part of the explanation for why XS effects differ from the longitudinal effects.
Answer: We believe we have offered several explanations for the differences between the results of both approaches: simultaneity bias, the fact that prevalence odds ratios and risk ratios/incidence rate ratios are different measures, subjects less likely to admit misuse in interviews in the longitudinal approach, and loss to follow-up. We would be happy to add any other explanation the reviewer might have.
Comment: When fitting a generalized linear model, you can fit either a marginal model (using GEE) or a conditional model (using random effects). Unlike with a linear model, the interpretation of these two is not the same in a generalized linear model (like logistic regression). Your cross-sectional (baseline data only) model estimates the marginal (population-averaged) effect of exposure on outcome. The corresponding longitudinal model would be a logistic regression fit using generalized estimating equations (GEE). The longitudinal model you fit is a logistic regression with random effects which estimates a conditional (individual-level) effect.[and the rest of the comment]. If you want to compare the XS and longitudinal approaches, you have to clarify what it is you are trying to compare. What you are currently showing is that a XS approach (which is, by necessity, a population-average approach) gives different results than an individual-level longitudinal approach. Again, this is not new information.
Answer: We are sorry we do not fully understand the interpretation of the reviewer. Our comparison was not that sophisticated in essence. We just wanted to show that researchers should be aware about drawing conclusions without paying attention to the design. Our exercise was not a statistical one, but a public health-oriented one. We plainly compared cross-sectional designs, which, in essence use prevalence of misuse and longitudinal (or prospective or cohort) designs, which, in essence, use incidence (either incidence rate or cumulative incidence). In the cross-sectional approach, at baseline, there is a certain number of subjects which present the outcome. We related this (or these) with exposure, being aware that the outcome might have preceded exposure, in which case causation is blurred. In the longitudinal design, we eliminated from the study those subjects who already presented the outcome at baseline, and followed the rest of the population to until they presented the outcome. This way we could calculate the odds ratio as an unbiased measure of relative risk as in any cohort study. We do not understand the distinction between population-level and individual-level analyses. Again, our paper, meant as a call for caution in the interpretation and not as a statistical demonstration, is targeted towards general practitioners or public health employees who engage in KAP studies without knowing much about the different designs. We could certainly use other models for our purpose. For instance a Poisson model or Cox model could have been used as well for the cohort study as they represent the staple analyses in epidemiology, as we have done in our previous paper (Mallah et al. Association of knowledge and attitudes with practices of misuse of tranquilizers: A cohort study in Spain. Drug Alcohol Depend 2021; 225:108793). GEE could be another alternative. By calculating odds ratio using GLMM we aimed to illustrate the difference of OR calculated in 2 different approaches.
Comment: How were the 890 people you approached selected from the population? Specify. Sounds like you have a convenience sample? Add to the Discussion a statement about how your sampling method results in a non-representative sample and your results may be biased.
Answer: The 890 people were selected randomly from adult children’s caregivers attending a large university hospital. They were not meant to be representative of the general population as our objective was not to determine the prevalence of misuse but rather to assess the relation of misuse with some knowledge and attitude independent variables. Cohort studies are not carried out on representative populations. For instance, the two most productive cohort studies in the world, the Nurses Health Study and the Physician Health Study, are, as their name indicates, carried out on American physicians and nurses, hence on a sample that is by no means representative of any population. Nonetheless, for decades, these cohort studies are the source on epidemiologic knowledge. As previously confirmed by methodology experts, representativeness of the sample is not necessary and should even be avoided (Rothman KJ,Gallacher JE, Hatch EE. Why representativeness should be avoided. Int J Epidemiol 2013;42:1012–1014).
Comment: Add the # of responses at each follow-up.
Answer: This part of the results was already published in the recent paper that presented the complete cohort study (Mallah et al. Association of knowledge and attitudes with practices of misuse of tranquilizers: A cohort study in Spain. Drug Alcohol Depend 2021; 225:108793). We believe that reproducing results from another paper is redundant. However, we would be willing to present these results if the reviewer confirms that they are necessary.
Comment: Clarify which of Q1 - Q16 are K and which are A. That would help in reading the Results.
We added the following description to the third paragraph of section 2.1: “Questions 1 to 4 and 11 and 13 explored personal Attitudes towards tranquilizers. Questions 5 to 8 and 10 determined participants´ Knowledge about tranquilizers. Questions 9 and 12 to 16 investigated patients´ attitudes towards healthcare provider (Mallah, Rodriguez-Can, et al., 2021).”
Comment: Table 1. Add a column for "No Misuse".
Answer: The column for no misuse is the difference between the first (total) and second column (misuse). We sincerely believe that adding such a column will not provide any additional information, however we may add it if the reviewer thinks we should.
Comment: Clarify the levels of "Educational level". You could have <HS, HS but no university, some university but not graduated, graduated from university. What do your levels include?
Answer: We did not ask subjects whether they graduated or not. In Spain, in general, people who claim they have university level mean they have a degree. We do not think that such a detail in this question has any importance. Our questionnaire asked about different educational levels, but the city in which the study took place is a university city. This is why we preferred to present our results as university level / below university level
Comment: Clarify "Family size". Who does this include?
Answer: Family size is the response to the question “How many members live in your house, including you?” included in the questionnaire. The questionnaire has been published elsewhere (Mallah N, et al. Development and validation of a knowledge, attitude and practice questionnaire of personal use of tranquilizers. Drug Alcohol Depend. 2021;224:108730).
Comment: Table 2. The header makes it sound like your Longitudinal sample of 1343 is just follow up data, not the baseline data. That makes sense since that way the exposures (which were measured only at baseline) all precede your outcome. If you actually included the baseline outcomes in the longitudinal dataset, take them out so your exposures all precede your outcomes. Clarify in your Methods that the longitudinal data does not include baseline outcomes, and that that allows causal inferences.
Answer: We added this comment to the methods section “To allow for causal inference, the longitudinal data did not include baseline outcomes”.
Comment: "Variables that showed a p-value <0.2 were introduced successively into the multivariate model, and those that modified the OR previously adjusted for age and gender by 10% or more were included in the final model (Greenland, 1989). Only outcomes of sufficient observations at baseline and during the follow-up were evaluated. "Using p-values to decide which confounders to include in your adjusted model biases your final model p-value. Instead, decide ahead of time which potential confounders to include and include them regardless of significance or magnitude of OR. See the text "Regression Modeling Strategies" by Frank Harrell, Jr. for more information. Greenland 1989 which you cite states, "variable-selection algorithms in current packaged programs, such as conventional stepwise regression, can easily lead to invalid estimates and tests of effect". What you did (p < .2 as your p to enter) is essentially the same thing as a stepwise procedure. But even the 10% rule can be problematic. Deciding what confounders to include based on the strength or magnitude of association of the confounders with the outcome leads to invalid p-values. Instead, decide a priori what variables may be confounders (excluding those that may be mediators) and include them all in the model, with the caveat that you should not have more than m/10 predictors in the model, where m = min(# with Y = 1, # with Y = 0). (Harrell, p73).
Answer: We would like to clarify that we chose our potential confounders based on an extensive literature review and previous knowledge on which variables could be included. The first selection of potential confounders was done even before developing the questionnaire that we used in this study. From this list of potential variables, we screened those that could be candidates. A criteria was the tentative point of p=0.2. As in any modeling situation, our intention was to come up with the most parsimonious model. This is by no means a stepwise modeling as it is not based on mechanical procedures in which the investigator does not have control on the data. We believe that introduction of confounder based on educated guess or on previous studies carried out elsewhere is not sufficient. If hundreds of confounders have been introduced in former studies taken altogether, we do not believe it is sound to introduce all of them in our model, without any screening. Indeed, confounding is a feature of the data at hand, i.e. a variable can be a confounder in our database but not in that of the neighboring study. Confounder is different from effect modification which is a universal feature. Whether a modeling strategy is better than other is a discussion that is beyond a simple paper such as ours. We have used the most frequently and knowledgeable method of confounding assessment in epidemiology.
Comment: You carried out a large number of tests and did not adjust for multiple testing. Thus, your Type I error rate is far larger than 0.05. In the Limitations, note that you did not adjust for multiple testing and these results should be considered exploratory (hypothesis generating), not confirmatory. If in future for a different study you want to do a confirmatory analysis, specify a set of primary tests a priori and adjust for multiple testing over those tests, with everything else considered secondary. But you can't choose based on the p-values... you would have had to choose ahead of time. So for this analysis you are limited at this point to keeping this as exploratory.
Answer: Except for the screening of confounders, using the very large 0.2 p-value, we did not carry out any hypothesis test. We calculated confidence intervals instead. And those concepts are very different as far as one does not use the 0.05 value as a cut-off between significant and non-significant. Adjusting for multiple comparisons is a topic that has been debated back in the 1990’s. Together with methodologic experts such as Ken Rothman, we believe that no adjustment should be made in that case. Rothman in his essay “No adjustments are needed for multiple comparisons. Epidemiology 1990; 1: 43-46” stated: “Adjustments for making multiple comparisons in large bodies of data are recommended to avoid rejecting the null hypothesis too readily. Unfortunately, reducing the type I error for null associations increases the type II error for those associations that are not null. The theoretical basis for advocating a routine adjustment for multiple comparisons is the “universal null hypothesis” that “chance” serves as the first-order explanation for observed phenomena. This hypothesis undermines the basic premises of empirical research, which holds that nature hollows regular laws that may he studied through observations. A policy of not making adjustments for multiple comparisons is preferable because it will lead to fewer errors of interpretation when the data under evaluation are not random numbers but actual observations on nature.”
In addition, we would like to clarify that association of each Knowledge or Attitude statement with tranquilizer misuse was independently assessed in a separate model.
Comment: I found the Results section difficult to read. I went into it thinking your main point was comparing XS and longitudinal effects, but you mix between highlighting interesting associations and comparing these effects.
Answer: We have considerably reduced the results section and just mention the differences between results of the two approaches.
Comment: "Moreover, cross-sectional and longitudinal designs evaluate different epidemiological concepts and readers could misinterpret a prevalence odds ratio of cross-sectional studies as a proxy of relative risk determined by longitudinal studies (Rothman, 1986)." But you did not compute relative risk in the longitudinal model.
Answer: We meant the following: When an OR is used, for instance in a case-control study, it is an unbiased estimate of the Incidence Rate Ratio. But when a Prevalence Odds Ratio is used in a cross-sectional study, it is not an unbiased estimate of the Incidence Rate Ratio.
Comment: "Nevertheless, in our study, the level of Knowledge and Attitudes of participants lost to follow-up was similar to those included in the study" This is not clear. What was your response rate at each follow-up?
Answer: We meant that those subjects who participated in the follow-up until the occurrence of the outcome or the end of the study did not have levels of knowledge that were different from those of subjects who abandoned follow-up. We wanted to show that there was no bias due to differential follow-up.
Reviewer 2 Report
This article tackles an important topic of tranquilizers misuse. The authors use appropriate English and apply accurate structure to present other papers regarding this topic as well as compare their findings with previous reports. Nevertheless, I have serious considerations regarding presentation of results. While I consider Tables 1 and 2 suitable additional to the results section, Tables 3 and 4 need improvement.
These two Tables contain too many variables and are impossible to interpret. It is also not clear what was authors intention behind depicting each item separately. I recommend presented results in more compact way so that it stays in line with the main message of the article or moving these two Tables (3 and 4) to the Supplement. Similarly, I suggest removing ORs from the sections 3.3 and 3.4 and leave reference to the Supplementary Tables. When reading this section it is very confusing what are the main results and what the authors want to convey.
Author Response
Reviewer #2:
Comment: These two Tables contain too many variables and are impossible to interpret. It is also not clear what was authors intention behind depicting each item separately. I recommend presented results in more compact way so that it stays in line with the main message of the article or moving these two Tables (3 and 4) to the Supplement.
Answer: Following this suggestion, we moved table 3 and 4 to the Supplement.
Comment: Similarly, I suggest removing ORs from the sections 3.3 and 3.4 and leave reference to the Supplementary Tables.
Answer: We have eliminated the numerical results and we now refer to the supplemental tables only.
Round 2
Reviewer 1 Report
Thank you for the thorough response to the review and for clarifying many aspects of the paper both in the text and in your response.
Minor revisions:
Table 1:
- The third column of Table 1 has the header "(N=75)". Change to "Misuse (N-75)".
- To help the reader directly compare the two groups without having to do their own calculations, I suggest adding a column for "No Misuse".
Section 2.1:
- Q13 is included in both Attitude toward tranquilizers and Attitude toward Healthcare provider. If intentional, no problem. If not, clarify.
Section 3.2:
- Should the reference to "Table 3" be to "Supplementary Table 1"?
- Should the reference to "Table 4" be to "Supplementary Table 2"?
Sections 3.3 and 3.4:
- "Q5-Q8 and Q10" are listed as the relevant variables in Section 3.2. Could you list the specific variables in 3.3 and 3.4, as well, to make it easier for the reader to know where to look in the tables? (These are listed in 2.1 but it would be helpful to have them in 3.3 and 3.4, as well, to make it easier for the reader).
Section 3.4:
- "Discrepancies in the results of the two approaches were more frequent in the case of patients´ Attitudes towards healthcare provider than in the case of personal Attitude towards tranquilizers."
If possible, could you clarify what discrepancies were more frequent? I found it hard to verify this conclusion from the tables. Number of times the CI contains 1? Those don't not seem very different in frequency. Magnitude of difference/ratio of ORs? Hard to tell that one group has more larger differences than the other.
Optional:
- To clarify a previous comment, when you use a GLMM with binary data the OR assesses how the odds would change for an individual if their attitude or knowledge were to change. When you use XS, the OR assesses how the odds differ between groups of individuals with different attitude or knowledge. The GEE approach uses longitudinal data to estimate the same thing as XS - how the odds differ between groups of individuals. So GLMM and XS (and GEE) are estimating different things (you already point that out in the discussion with respect to XS vs. GLMM).
See, for example:
https://bmcmedresmethodol.biomedcentral.com/articles/10.1186/1471-2288-2-15
https://www.ncbi.nlm.nih.gov/pmc/articles/PMC4217193/
So, technically, you are comparing "GLMM" vs. XS, not "longitudinal" vs. XS (since longitudinal includes both GLMM and GEE). You encourage readers to use a longitudinal approach, which might then lead someone to use GEE (when they look up longitudinal methods, they might come across GEE even if you don't mention it) when in fact they want to use GLMM. Some mention of the GLMM vs. GEE distinction could be added to the Discussion if you think it would help with the goal of your paper.
Author Response
Dear Dr. Li,
We would like to thank you and the reviewer for reviewing our manuscript.
Below we provide our response to each of the comments made by the reviewer (review report round 2). We also provide a version of the article in which all changes are tracked.
We hope that the amended paper will now be deemed acceptable for publication in the Journal of Clinical Medicine.
Sincerely,
Bahi Takkouche MD, Ph.D
Professor of Epidemiology
University of Santiago de Compostela
Spain
Reviewer 1.
Minor revisions:
Comment #1. Table 1:
A) The third column of Table 1 has the header "(N=75)". Change to "Misuse (N-75)".
Answer: We added the header “non-medical use” to this column, as the editor had asked us in a previous review round to replace the term “misuse” by “non-medical use”.
- B) To help the reader directly compare the two groups without having to do their own calculations, I suggest adding a column for "No Misuse".
Answer: We added the column “no non-medical use”. Kindly check the updated table 1.
Comment #2. Section 2.1:
- Q13 is included in both Attitude toward tranquilizers and Attitude toward Healthcare provider. If intentional, no problem. If not, clarify.
Answer: We would like to clarify that the construct validation of the used questionnaire indicated that item Q13 shares content with the two Attitude factors (Attitude toward tranquilizers and Attitude toward Healthcare provider). To avoid any confusion, we now say in L185-187 of the updated manuscript:
“The item Q13 shares content with the two Attitude factors (Attitude toward tranquilizers and Attitude toward Healthcare provider) (Mallah, Rodriguez-Can, et al., 2021).”
Comment #3. Section 3.2:
A) Should the reference to "Table 3" be to "Supplementary Table 1"?
Answer: We replaced “Table 3” by “Supplementary Table 1” in L304 of the updated version of the manuscript.
B) Should the reference to "Table 4" be to "Supplementary Table 2"?
Answer: We replaced “Table 4” by “Supplementary Table 2” in L310 of the updated version of the manuscript.
Comment #4. Sections 3.3 and 3.4:
- "Q5-Q8 and Q10" are listed as the relevant variables in Section 3.2. Could you list the specific variables in 3.3 and 3.4, as well, to make it easier for the reader to know where to look in the tables? (These are listed in 2.1 but it would be helpful to have them in 3.3 and 3.4, as well, to make it easier for the reader).
Answer: In the updated version of the manuscript, we specified the variables in sections 3.3. and 3.4 as requested by the reviewer. We now say:
L314 “…personal Attitudes towards tranquilizers (Q1-Q4, Q11 and Q13)…”
L320-321 “…Attitudes towards healthcare provider (Q9 and Q12-16)...”
Comment #5. Section 3.4:
- "Discrepancies in the results of the two approaches were more frequent in the case of patients´ Attitudes towards healthcare provider than in the case of personal Attitude towards tranquilizers."
If possible, could you clarify what discrepancies were more frequent? I found it hard to verify this conclusion from the tables. Number of times the CI contains 1? Those don't not seem very different in frequency. Magnitude of difference/ratio of ORs? Hard to tell that one group has more larger differences than the other.
Answer: We are sorry if we were not clear enough in this paragraph. We have updated it as follows:
L319-327: “There are discrepancies in the results of the two approaches regarding the association of patients´ Attitudes towards healthcare provider (Q9 and Q12-16) with non-medical tranquilizer use. In particular, item Q12 (incomplete trust with the doctors’ decision to prescribe tranquilizers or not) was not associated with the outcome “any non-medical tranquilizer use practice” in the cross-sectional approach, contradicting the results of the longitudinal approach (Supplementary Table 1). Nonetheless, item Q12 was associated with higher odds of the outcomes “shortening the course of treatment” and “modifying the prescribed dose”, whereas in the longitudinal approach, no association was observed between Q12 and “modifying the prescribed dose” (Supplementary Table 2).”
Comment #6. Optional:
- To clarify a previous comment, when you use a GLMM with binary data the OR assesses how the odds would change for an individual if their attitude or knowledge were to change. When you use XS, the OR assesses how the odds differ between groups of individuals with different attitude or knowledge. The GEE approach uses longitudinal data to estimate the same thing as XS - how the odds differ between groups of individuals. So GLMM and XS (and GEE) are estimating different things (you already point that out in the discussion with respect to XS vs. GLMM).
See, for example:
https://bmcmedresmethodol.biomedcentral.com/articles/10.1186/1471-2288-2-15
https://www.ncbi.nlm.nih.gov/pmc/articles/PMC4217193/
So, technically, you are comparing "GLMM" vs. XS, not "longitudinal" vs. XS (since longitudinal includes both GLMM and GEE). You encourage readers to use a longitudinal approach, which might then lead someone to use GEE (when they look up longitudinal methods, they might come across GEE even if you don't mention it) when in fact they want to use GLMM. Some mention of the GLMM vs. GEE distinction could be added to the Discussion if you think it would help with the goal of your paper.
Answer: We thank the reviewer for the clarification and for sharing those interesting papers. If the reviewer does not mind, we prefer not to add technical discussions to the manuscript as it is mainly oriented to clinicians and public health researchers who most often do not have a deep statistical background. In the manuscript, we stated that we adopted GLMM approach to explain the undertaken analysis, but we prefer to keep the manuscript simple in that regard to facilitate the comprehension of the main message by the reader.